# Efficacy and safety of combined oral sucrose and nonnutritive sucking in pain management for infants: A systematic review and meta-analysis

Qiaohong Li[1☯], Xuerong Tan[1☯], Xueqing Li[2], Wenxiu Tang[1], Lin Mei[1], Gang Cheng[1], Yongrong Zou[1]*

1 Department of Neonatology, Ya'an People's Hospital, Ya'an, Sichuan, China, 2 Department of Pediatrics, Ya'an People's Hospital, Ya'an, Sichuan, China

☯ These authors contributed equally to this work.
* zwj135678@126.com

**Funding:** Our paper is supported by Project of Sichuan Provincial Health Commission in 2014 (NO: 140108). The funders had no role in study

## Abstract

### Background

Pain management is currently important in neonatal intensive care unit (NICU). The superiority in pain relief of the combined oral sucrose (OS) and nonnutritive sucking (NNS) to other single intervention has not been well established. The administration of sucrose has been considered to potentially induce adverse events, which has been controversial. This study aims to investigate the combined effects and safety in comparison with other single intervention methods, including NNS, OS alone, breast milk and oral glucose.

### Methods

We searched databases including Medline (via Pubmed), Embase (via Ovid), web of science, and Cochrane Library for randomized controlled trials from Jan 1, 2000 to Mar 31, 2021. The data were analyzed in the meta-analysis using Review manager Version 5.3. Pain score was the primary outcome in this meta-analysis. The adverse events were assessed qualitatively.

### Results

A total of 16 studies were eligible in the meta-analysis. The results demonstrated a significant reduction in pain score in the NNS+OS group compared with NNS alone (SMD = -1.69, 95%CI, -1.69,-0.65) or sucrose alone (SMD = -1.39, 95% CI, -2.21,-0.57) during the painful procedures. When compared NNS+OS with breast milk, no significant difference was detected (SMD = -0.19, 95% CI: -0.5, 0.11).

### Conclusion

The combined effects of NNS and OS might be superior to other single intervention method. However, the effects might be mild for moderate-to-severe pain.

design, data collection and analysis, decision to publish, or preparation of the manuscript.

**Competing interests:** The authors have declared that no competing interests exist.

**Abbreviations:** NICU, neonatal intensive care unit; OS, oral sucrose; NNS, nonnutritive sucking; PIPP, Premature Infant Pain Profile; PIPP, R: premature infant pain profile-revised; NIPS, Neonatal Infant Pain Scale; NFCS, Neonatal Facial Coding System; N-PASS, Neonatal Pain Agitation and Sedation Scale; WMD, weighted mean difference; SMD, standardized mean difference; ROP, retinopathy of prematurity; SDs, standard deviations.

# 1 Introduction

Newborns, especially preterm infants, are frequently subjected to painful procedures [1], repeated painful and stressful stimuli may develop clinical, physiologic and psychologic sequelae in the short or long term [2–4]. In some developing countries, most infants were not provided any analgesic intervention during painful procedures [5, 6]. Even in a developed country like Canada, analgesic interventions were not offered in almost half of painful procedures, according to an epidemiology study [7]. In addition, it remains unclear whether the administration of anesthetics is safe [8]; thus, selective use of pharmacological treatment is recommended, such as opioids, non-opioids, and other anesthetics [9]. On the other hand, non-pharmacological methods have been developed in recent years to help reduce multiple bedside interruptions and have been considered safe and effective in pain management [10, 11]. An updated research revealed the use rate of nonpharmacologic interventions tended to be higher than pharmacologic interventions [12].

Combined therapy was assumed to be more efficient in pain relief than single interventions and has been investigated frequently in recent studies [13–16]. For example, combined effects of music and touch [17], sucking, breast milk and tucking [18], music and sucrose [19], oral sucrose (OS) and nonnutritive sucking (NNS) [20, 21], etc. were evaluated in clinical trials. Among which the combined effects of NNS and sucrose were tested by numerous prospective randomized trials. It has been common knowledge that OS and NNS could separately alleviated pain in neonates. Previous reviews or meta-analyses have preliminarily evaluated the combined effects of NNS and sucrose [10, 13]. However, they may have neglected the variations in effectiveness and safety under different painful procedures. The superiority of the combination to various single interventions was also barely covered [8]. Therefore, this systematic review and meta-analysis aim to fill the gap by evaluating the efficacy of combining OS and NNS in different scenarios of comparison and exploring the safety of this intervention method preliminarily.

# 2 Methods

## 2.1 Literature search and screen

We conducted a systematic search from the following databases for English language articles: Medline (via PubMed), Embase (via Ovid), Cochrane Library, Web of Science. The search terms were based on three domains "newborns", "sucrose and non-nutritive sucking" and "procedural pain". The outcome-related terms were not restricted. Randomized controlled trials were selected using database-specific limiters. The time span of publication years was limited from Jan 1, 2000 to Mar 31, 2021. The search strategies for each database were provided (S1 Table). In addition to the electronic searches, Google Scholar and references in literature were also manually searched for potential suitable studies. The unpublished studies were not considered. The language restriction was English.

The literature research and screening procedures afterwards were performed independently by two researchers. The final eligible studies were cross-checked. If there were any disagreements, a third author made judgments.

## 2.2 Inclusion and exclusion criteria

**2.2.1 Inclusion criteria.** (1) The study was a randomized controlled trial. (2) The paticipants were preterm or full term neonates without severe illness. (3) The randomized groups in the trial should at least contain an intervention group applied with OS combined with NNS. (4) Conference abstracts eligible for the inclusion criteria above were also included.

**2.2.2 Exclusion criteria.** (1) Results of pain score were presented as categorical outcome. (2) The sample size, mean and standard deviation were partly provided and could not be estimated from other statistics such as mean difference and p-value or median, range and interquartile range. (3) Non-English literature was excluded. (4) The outcomes of no interest, e.g., the occurrence of startle, jerk or tremor were excluded.

## 2.3 Literature quality evaluation

We conducted literature quality evaluation according to the Cochrane Risk of Bias Tool (RoB), version 5.2 [22]. The risk of bias was assessed on a 3 level scale:"low", "high" and "unclear" risk based on the following seven domains: random sequence generation, allocation concealment, blinding of participants and personnel, blinding of outcome assessment, incomplete outcome data, selective reporting, intention-to-treat analysis, and a completeness of follow-up.

## 2.4 Data extraction

Data extraction was performed by two reviewers independently. The summary of studies was established from the following information: first author, year of publication, country, study design, gestational age, pain procedure, sample size and specific intervention descriptions in each group and outcome.

We implemented the meta-analysis on pain score measured in different painful procedures, including Premature Infant Pain Profile(PIPP) [23], the premature infant pain profile-revised (PIPP-R) [24], Neonatal Infant Pain Scale (NIPS) [25], the Neonatal Facial Coding System (NFCS) [26] Neonatal Pain Agitation and Sedation Scale (N-PASS) [27]. The results of pain score were required to be presented as continuous variables. The Means and standard deviations (SDs) were extracted directly from the original data if available. Otherwise, they were estimated using Hozo et al.'s Method or Bland's Method under the scenario where median, IQR or ranges were available [28–30]. Calculations using p-value, confidence interval and mean difference were performed following the instructions in Cochrane handbook 5.1 [22]. Statistics were estimated from figures if applicable when no digits were provided. Missing data were not imputed. For cross-over designs trials, paired analysis was performed to standardize the means and SDs to account for the within-subject correlation [31]. If no information about within-subject correlation was provided, we decided to assume the correlation to be 0 which was a conservative way since there were no similar data to be referred in the included studies.

For evaluating the effects at different timepoints, we extracted the outcome data in 2 phases, (1) during the procedure phase (during or within the first minute immediately after the painful procedure), and (2) the recovery phase (1 to 5 min after the procedure). Pairwise comparisons were performed between the intervention group of interest, i.e., sucrose and NNS group and any other relevant groups including NNS (NNS+water)/sucrose alone, breastfeeding, breast milk, glucose or routine care group. In crossover trials, the overall effects were extracted instead of looking at a single sequence.

## 2.5 Statistical analysis

Results of pain score were presented as standardized mean difference (SMD) if holding different scales and 95% CI considering variety among pain measurements. If studies shared the same measurement then mean difference (MD) could be presented. A p-value <0.05 was considered as significant. Statistical heterogeneity was assessed using $I^2$ tests. Subgroup analysis or random-effects model applied when the statistical heterogeneity was high ($I^2 > 50\%$) [22]. Otherwise, a fixed-effects model was conducted. Sensitivity analyses were conducted to assess the stability and validity of results. Factors were considered when removing certain studies,

including article types, outcome measurements (PIPP or non-PIPP), target population (preterm/full term), etc. Publication bias was evaluated by Egger's Test and funnel plots. All the statistical analysis was performed using Review Manager 5.3 [32]. Egger's Test was performed using SAS version 9.4. Besides, the safety of the intervention was qualitatively evaluated based on the occurrence and types of adverse events.

# 3 Results

## 3.1 Outline of eligible studies

A total of 232 records were identified using our searching strategy from databases and two from the references in a previous review [10]. After removing duplicates, 79 records were obtained and 63 of them were screened in the first stage by looking through abstracts and titles. The first round of screening led to 31 records for the following full-text review. Finally, 16 studies [33–40, 42–49] were included in our systematic review and meta-analysis. The screening process was described as the PRISMA flow diagram (Fig 1).

There were three conference abstracts among included studies [33–35]. Of all the RCTs included, two were based on a cross-over design [34, 36]. Eligible Infants enrolled were categorized by preterm (n = 8), full-term (n = 5) and both (n = 3). Infants underwent different

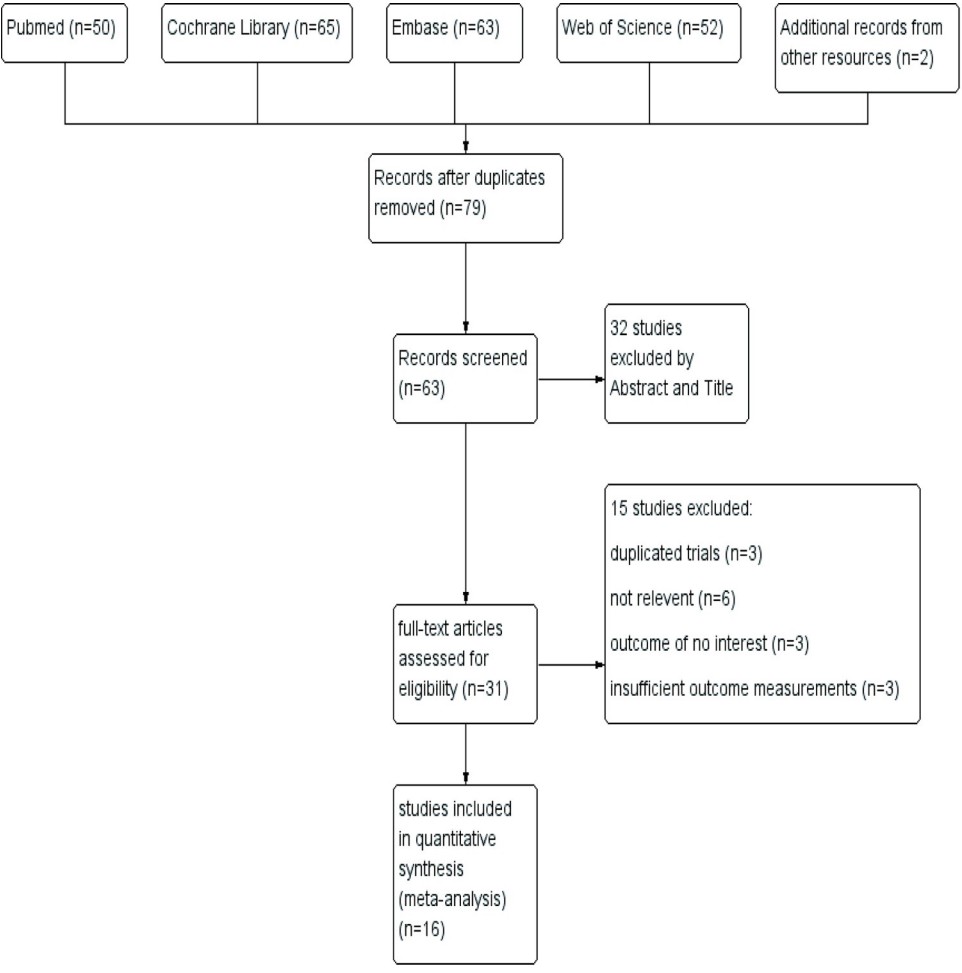

**Fig 1. Flow diagram of study inclusion and exclusion.**

painful procedures including heel stick (n = 8), retinopathy of prematurity (ROP) screening (n = 5), venipuncture (n = 2) and wound dressing (n = 1). PIPP score was mostly widely used for pain assessment (n = 10) while PIPP-R(n = 1), NIPS (n = 3), NFCS (n = 1) and N-pass score(n = 1) were also applied in other studies. Characteristics of eligible studies were outlined in Table 1. Fig 2A, 2B demonstrated the risk of bias assessment for each study.

**Table 1. The characteristics of included studies.**

| First author, year, country | Study Type/ design | Objects | Procedure | Sample size | Gestational age and weight (inclusion criteria) | Intervention/Control groups | Outcomes |
|---|---|---|---|---|---|---|---|
| Asmerom,2013, United States [46] | Prospective double-blind randomized controlled study | preterm | heel lance | sucrose+NNS group: n = 44 | premature infants ≤ 36.5 | sucrose+NNS group: | PIPP scored from the time of heel lance to 30 seconds post the lance. |
| | | | | water+NNS group: n = 45 | weeks gestation who weighed ≥800 grams | received a single dose of 24% sucrose via syringe to the anterior tongue along with a pacifier (NNS) two minutes before the heel lance. | |
| | | | | | | Water+NNS group: the placebo group received an equal volume of sterile water to the anterior portion of the tongue along with a pacifier. | |
| Benoit,2021, Canada [47] | Single-blind randomized controlled trial | healthy full-term | heel lance | sucrose+NNS: n = 19 | healthy, full-term (born ≥ 37 0/ 7 weeks'gestation) | Intervention group: 24% oral sucrose combined with offered NNS and containment in a blanket while in an infant cot. Control group: direct breastfeeding | 1.pain-related brain activity |
| | | | | breastfeeding: n = 18 | | | 2.PIPP-R at 30-, 60-, 90-, and 120-s following heel lance |
| | | | | | | | 3.adverse events |
| Boyle, 2016, Canada [42] | Prospective, Randomised, placebo controlled study | preterm | ROP screening | group 1: n = 10 | < 32 weeks' PMA | Group 1: 1 ml sterile water given by mouth using a syringe | PIPP scores during eye examination |
| | | | | group 2: n = 10 | | Group 2: 1 ml sucrose 33% given by mouth using a syringe | |
| | | | | group 3: n = 9 | | Group 3: 1 ml sterile water given by mouth using a syringe and pacifier put into the mouth | |
| | | | | group 4: n = 11 | | Group 4: 1 ml sucrose 33% given by mouth using a syringe and pacifier put into the mouth | |
| Collados,2018, Spain [36] | Multicentre randomised, non-inferiority, cross-over trial | preterm | venipuncture | EBM-sucrose sequence: n = 33 | gestational age of less than 37 weeks at birth and weigh less than 2,500 grams. | In group one, the neonate received EBM during the first venipuncture that was included in the study and 24% sucrose during the second venipuncture. The process was reversed for group two. | PIPP, the duration of crying, oxygen saturation and heart rate |
| | | | | sucrose-EBM sequence: n = 33 | | | |
| | | | | | | This accompanied throughout by non-nutritive sucking and swaddling. | |
| Cullas,2012, Turkey [33] | Prospective randomised study | preterm | ROP screening | sucrose+NNS: n = 21 | patients under 32 weeks of gestational age | Group 1: oral sucrose solution given two minutes before examination. Pacifier was used. | PIPP, time of crying |
| | | | | water+NNS: n = 19 | | Group 2: sterile water given two minutes before examination. Pacifier was used. | |
| De Bernardo,2019, Italy [48] | Randomized double-blinded case–control pilot study | full-term | venipuncture | sucrose+NNS: n = 33 | neonates 37–42 weeks gestational age at birth and >1 week old at the time of the intervention with body weight 2,500–4,500 g and able to feed orally. | Intervention group: received both 1 mL 24% sucrose orally via syringe 1 minute before venipuncture and 1 mL during the procedure. A pacifier was offered to all neonates immediately following sucrose administration each infant. | NIPS |
| | | | | glucose+NNS: n = 33 | | | Outcome measurements (HR, SpO2) were obtained before (T0), during (T1), and 1 minute after (T2) venipuncture |
| | | | | | | Control: received 1 mL 10% glucose orally via syringe with a pacifier 1 minute before venipuncture and during the procedure | |

*(Continued)*

**Table 1.** (Continued)

| First author, year, country | Study Type/ design | Objects | Procedure | Sample size | Gestational age and weight (inclusion criteria) | Intervention/Control groups | Outcomes |
|---|---|---|---|---|---|---|---|
| Dilli, 2014, Turkey [49] | Prospective randomised and placebo-controlled study | preterm | ROP screening | Group 1: n = 32 | - | Group 1: 0.5 mL/kg of sucrose 24% was given by mouth using a syringe, and pacifier was placed in the mouth. | PIPP score during examination crying time |
| | | | | Group 2: n = 32 | | Group 2: 0.5 mL/kg of sterile water was given by mouth using a syringe, and pacifier was placed in the mouth. | |
| Gao,2018, China [44] | Randomized controlled trial | Preterm infants | heel stick | 1.NNS group: n = 22 | before 37 weeks of gestation | 1.NNS group: pacifier given in 2 minutes before, and throughout the recovery phase of the heel stick. | 1.PIPP scale in the blood collection phase (0-60s) and recovery phase (after 1 min) |
| | | | | | | 2. sucrose group: Sucrose 20% (0.2 mL/kg) was administered to the preterm infant's mouth by 1 ml syringe in 2 minutes before the heel stick procedure | 2.heart rate and oxygen saturation |
| | | | | | | 3.sucrose+NNS: Sucrose 20% (0.2 mL/kg) was administered to the preterm infant's mouth by 1 ml syringe in 2 minutes before the heel stick procedure and a pacifier was given until the recovery phase of the heel stick. | 3.the percentage of crying time respectively in the blood collection phase and recovery phase |
| | | | | 2.sucrose group: n = 21 | | 4. routine care group: received only routine comfort through gentle touch when he cried after the heel stick procedure. | |
| | | | | 3.NNS+sucrose group: n = 22 | | | |
| | | | | 4.routine care group: n = 21 | | | |
| Gibbins,2002, Canada [45] | Randomized controlled trial | Preterm/ term | heel lance | 1.sucrose+NNS: n = 64 | born between 27 and 43 weeks gestation | 1. sucrose+NNS group: received 0.5 ml of 24% sucrose via a syringe followed immediately by the insertion of a pacifier into the mouth. The pacifier was held in place as required 2 minutes before, during, and 5 minutes following the heel lance. | the PIPP are numerically scored on 30/60 seconds following an acute painful stimulus. |
| | | | | 2.sucrose alone: n = 62 | | 2. sucrose group: received 0.5 ml of 24% sucrose via a syringe. | |
| | | | | 3.water+NNS: n = 64 | | 3.water+NNS group: received 0.5 ml of sterile water via a syringe. No pacifier was offered. | |
| Leng,2016, China [43] | Prospective, multi-centred, randomized controlled clinical trial | full term | shallow or deep heel stick procedures | NS group: n = 167 | gestational age between 37 and 42 weeks at delivery; | Group S: 2 ml of 24% sucrose was administrated by syringe 2 min before the heelstick procedure. | NFCS score |
| | | | | S group: n = 176 | Birthweight between 2500 g and 4000 g; | Group NS: 2 ml of 24% sucrose was administrated by syringe 2 min before the heel stick procedure, and then a standard silicone newborn pacifier was placed into the infant's mouth until the end of the process. | |
| Miller, 2009, United States [34] | Repeated-measures crossover design. | preterm/ full term | heel stick | NNS+sucrose-no treatment sequence: n = 7 | between the ages of 32 | 1.In the treatment condition, infants were offered NNS with sucrose. | NIPS score, heart rate and oxygen saturation |
| | | | | no treatment-NNS+sucrose sequence: n = 7 | weeks to younger than or equal to 42 weeks | 2.In the control condition, infants were not offered any treatment | |

*(Continued)*

**Table 1.** (Continued)

| First author, year, country | Study Type/ design | Objects | Procedure | Sample size | Gestational age and weight (inclusion criteria) | Intervention/Control groups | Outcomes |
|---|---|---|---|---|---|---|---|
| Mandee,2020, Thailand [37] | Prospective randomized control trial | preterm/ full term | Wound dressing | sucrose+NNS group: n = 16 | - | Sucrose+NNS group: participants were first administered a dose of 24% sucrose and then were given the pacifier. | The NIPS scores were assessed at 30, 120, and 240 seconds |
| | | | | NNS group: n = 16 | | NNS group: participants were administered a pacifier while their wound dressing was performed. | from the commencement of the wound dressing, crying time |
| Mitchell,2016, United States [39] | Double-blind 2×2 factorial randomized controlled trial | full term | heel stick | sucrose+NNS group: n = 37 | healthy term infants between 37 and 42 weeks | 1.sucrose+NNS: received 1±0.1 ml of the 24% sucrose solution orally with pacifier at 2 ±0.5 minutes prior to the procedure 2.water+NNS: received 1 ±0.1 ml of sterile water with pacifier | PIPP, heart rate variability (HRV), and salivary cortisol |
| | | | | water+NNS group: n = 39 | | | |
| O'Sullivan,2010, Ireland [38] | Randomised placebo controlled study | preterm | ROP screening | sucrose+NNS group: n = 20 | < 1501 g | 1. sucrose+NNS: infants were swaddled and received 0.2 ml of sucrose 24% given by mouth using a syringe and a soother | N-PASS score |
| | | | | water+NNS group: n = 20 | < 32 Weeks gestation Infants | 2. water+NNS: infantsswaddled, and received 0.2 ml of sterile water given by mouth using a syringe and a soother. | |
| Thakkar,2016, India [40] | Randomized controlled trial | full term | heel-stick | group I (sucrose): n = 45 | (>37 weeks PMA), with birthweight > 2200 g | 1.group I received 30% sucrose solution by sterile syringe; | PIPP score, heart rate, oxygen saturation, duration of crying |
| | | | | group II (NNS): n = 45 | | 2.group II received NNS in which sterile gauze was held gently in neonate's mouth and the palate tickled to stimulate sucking; | |
| | | | | group III (sucrose+NNS): n = 45 | | 3.group III received both the interventions (sucrose and NNS); | |
| | | | | group IV: n = 45 | | 4.group IV received no intervention. | |
| Ucar S,2014, Turkey [35] | Randomised, controlled study | preterm | prematurity (ROP) screening | sucrose group: n = 27 | - | group 1 received 24% sucrose oral, | PIPP |
| | | | | sucrose+NNS: n = 27 | | group 2 received 24% sucrose with pacifier | |
| | | | | water+NNS: n = 27 | | group 3 received sterile water with pacifier. | |

ROP: retinopathy of prematurity; NNS: nonnutritive sucking; PIPP: Premature Infant Pain Profile; NFCS: Neonatal Facial Coding System; NIPS: Neonatal Infant Pain Scale

## 3.2 Pain score

**3.2.1 NNS+OS group versus NNS group.** A total of 11 studies with 677 participants explored the effect of OS combined with NNS compared with NNS alone during the painful procedures, most of which reported PIPP score, while one study reported NIPS [37] and one reported N-pass score [38]. In both heel-stick and ROP subgroups, a significant effect was observed under the random-effects model, with the standard mean difference being -1.59 (95%CI: -2.49, -0.68) and -1.05(95%CI: -1.56, -0.55), respectively. However, there was no significant difference in the wound dressing subgroup (Fig 3). A sensitivity analysis was conducted based on two aspects: removing conference abstracts and removing articles using non-PIPP measurements. The results remained significant after two conference abstracts were removed in the ROP group [33, 35]. Besides, unifying the measurements by removing two studies reporting NIPS or N-pass score did not change the results or reduce heterogeneity

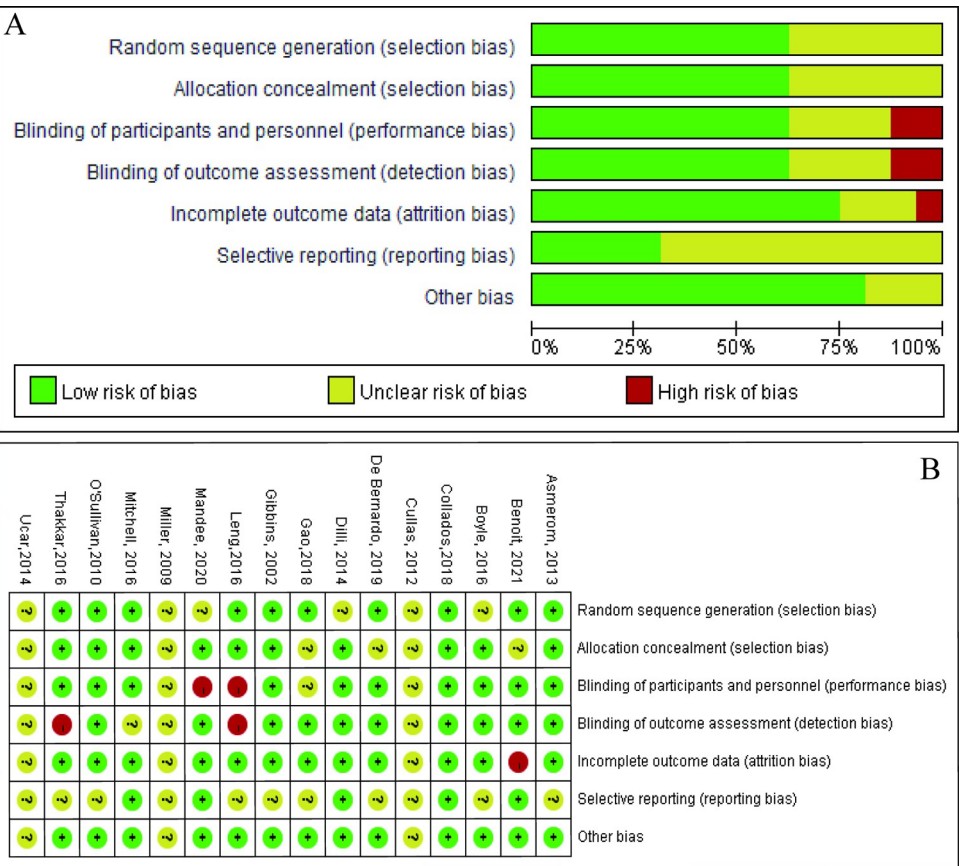

**Fig 2. Risk of bias assessment for each included trial.** A, Risk of bias graph; B: Risk of bias summary.

neither. The analysis showed no effect in four studies involving term infants regardless of the type of procedures (p = 0.16) with SMD: -0.81 (95%CI: -1.94, 0.33) [37, 39–41].

Fewer studies provided information in the recovery phase. The available 4 studies showed the combined sucrose and NNS did not provide a better effect in relieving pain in both heel stick and wound dressing groups (Fig 4). However, this finding was not robust in the heel stick subgroup due to the reverse result (MD = -3.23, 95%CI: -4.56, -1.89, P<0.001) after removing Mitchell's study in the sensitivity analysis [39].

**3.2.2. NNS+OS versus OS alone.** Six studies [35, 40, 42–45] involving 677 infants or newborns assessed the pain in the group applying sucrose alone. One study measured pain as NFCS score [43] while the other five studies reported PIPP. The meta-analysis showed a significant effect of the combined interventions than OS (Fig 5). No significant difference, however, was detected when studying the term subgroup (SMD = -1.30, 95%CI: -2.81, 0.22, P = 0.09) [40, 41, 43]. In the recovery phase, the effect was also significant (MD = -3.48, 95%CI: -5.41, -1.54) (Fig 6).

**3.2.3 NNS+OS versus routine care group.** Five studies involving 268 participants were included in the analysis to compare the effect between the NNS+sucrose group and the routine care group [34, 40, 42, 44, 46]. The results showed a better impact of NNS+sucrose in both heel stick (SMD = -2.66, 95%CI: -4.53, -0.78) and ROP group (SMD = -1.1, 95%CI: -2.03,-0.17) (Fig 7). Removal of any study did not change the results significantly.

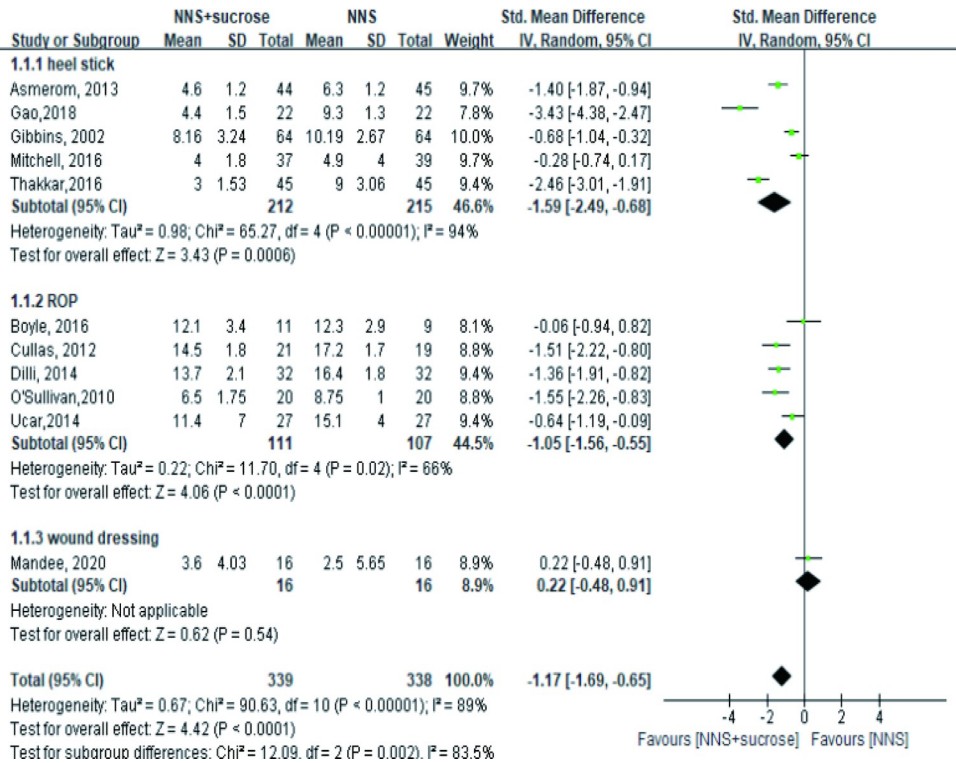

**Fig 3. Forest plot of pain score comparing the combined intervention (NNS and sucrose) with applying NNS alone during different painful procedures.** NNS: nonnutritive sucking.

**3.2.4 NNS+OS versus breastfeeding or breast milk.** The comparison between OS+NNS and breast milk showed no difference in effects for pain relief despite the heterogeneity in study design, intervention methods, pain procedures, outcome assessment and target

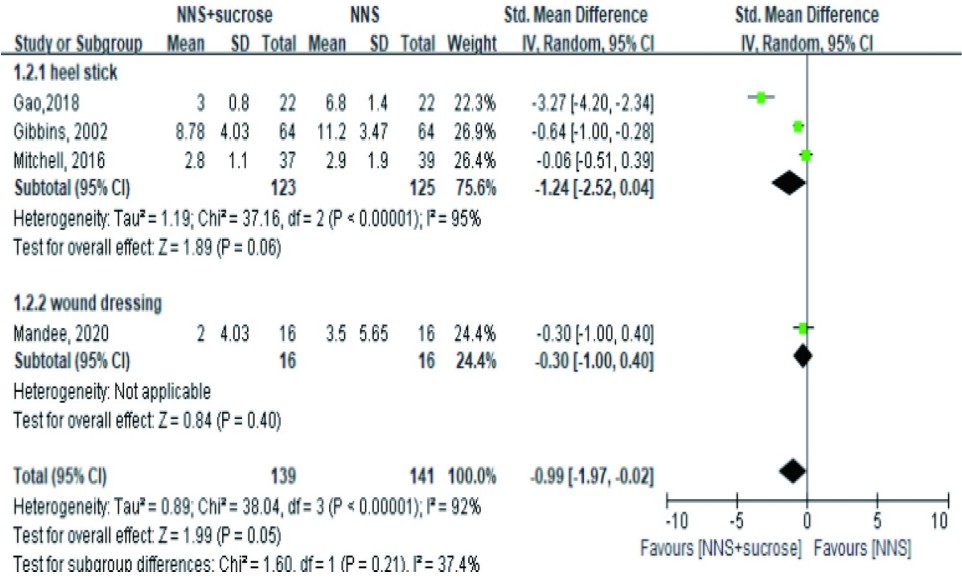

**Fig 4. Forest plot of pain score comparing the combined intervention (NNS and sucrose) with applying NNS alone in the recovery phase.** NNS: nonnutritive sucking.

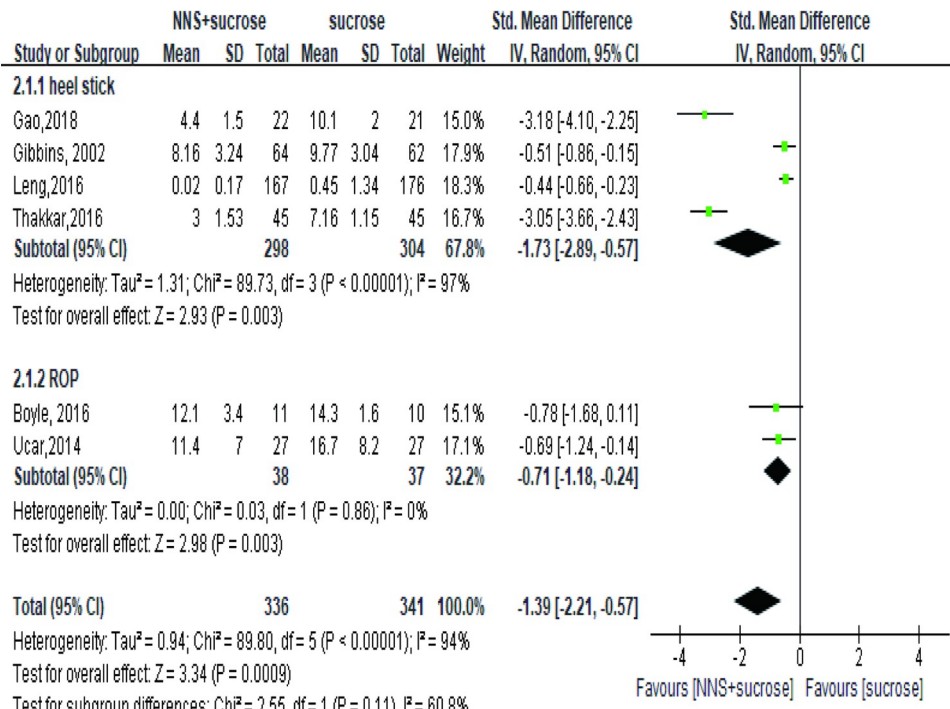

**Fig 5. Forest plot of pain score comparing the combined intervention (NNS and sucrose) with applying sucrose alone during heel stick and ROP.** NNS: nonnutritive sucking; ROP: retinopathy of prematurity.

population between the two studies (Fig 8). In Benoit's study [47], full term infants underwent heel stick and were direct breastfed, whose pain was assessed by PIPP-R. In contrast, 66 preterm infants underwent venipuncture in two sequences. During the controlling period, infants were given breast milk via a pacifier, whose pain was assessed by PIPP in Collados's study [36].

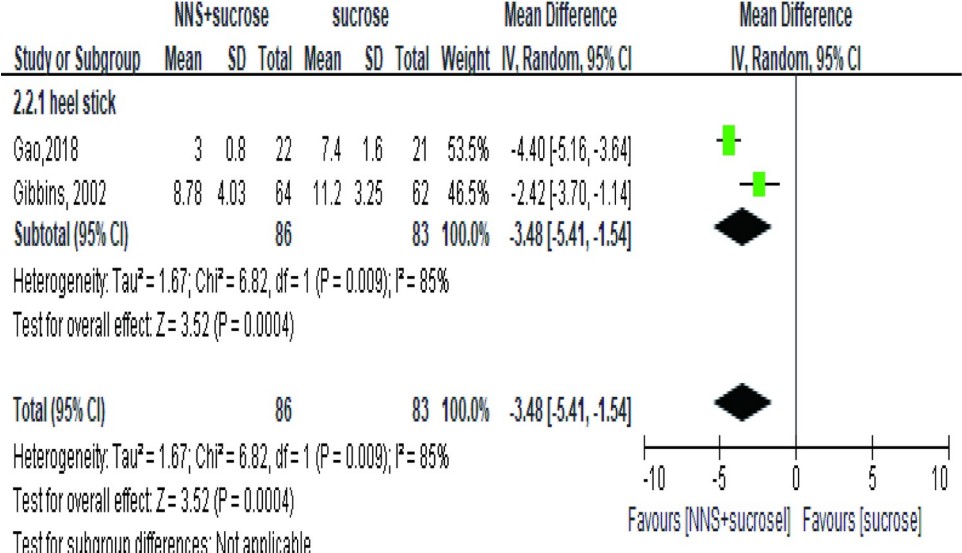

**Fig 6. Forest plot of pain score comparing the combined intervention (NNS and sucrose) with applying sucrose alone in recovery phase.** NNS: nonnutritive sucking.

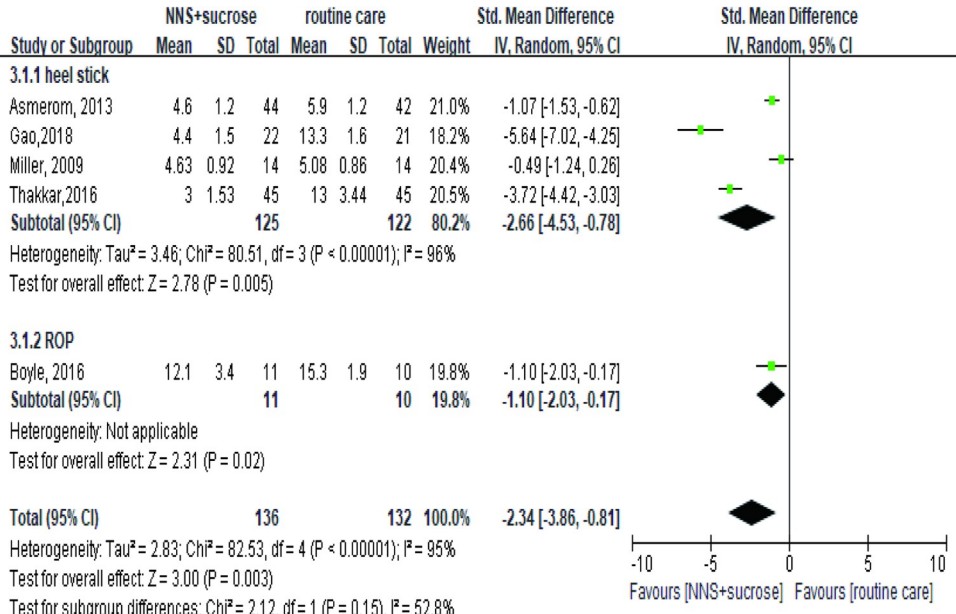

**Fig 7. Forest plot of pain score comparing the combined intervention (NNS and sucrose) with routine care group during the procedures.** NNS: nonnutritive sucking.

**3.2.5 NNS+OS versus glucose.** Only one study enrolling 66 term newborns set glucose +NNS as the control group [48]. The result demonstrated a significant difference (p < 0.05) in both two phases. NIPS scores were significantly lower in the NNS+sucrose group (range 1–2 and median 0) compared with the NNS+glucose group (range 5–7 and median 6) during venipuncture.

**3.2.6 Publication bias.** Publication bias was tested by Egger's Test and funnel plots (Fig 9A, 9B). The results of Egger's test indicated no publication bias in both scenarios. The bias

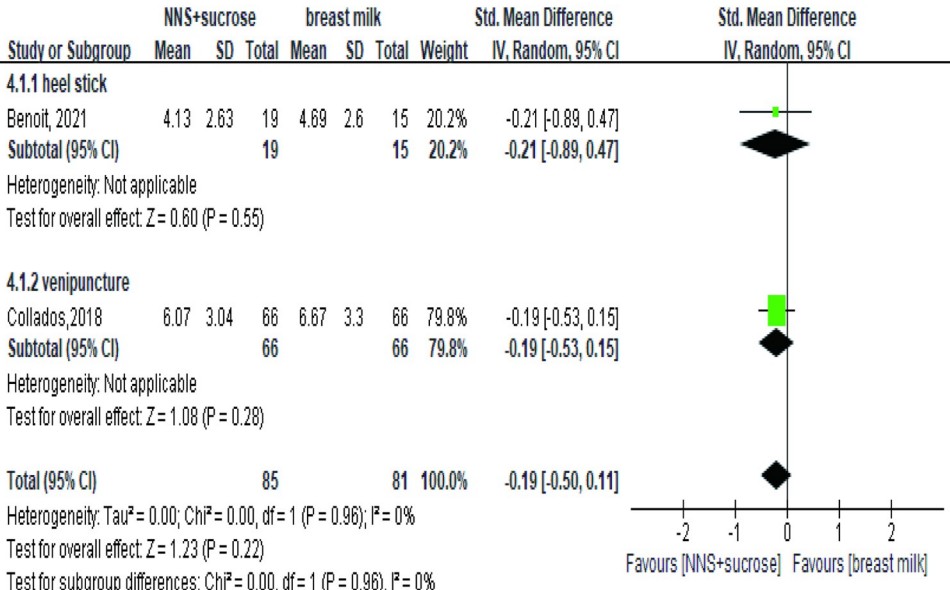

**Fig 8. Forest plot of pain score comparing the combined intervention (NNS and sucrose) with breastmilk during the procedures.** NNS: nonnutritive sucking.

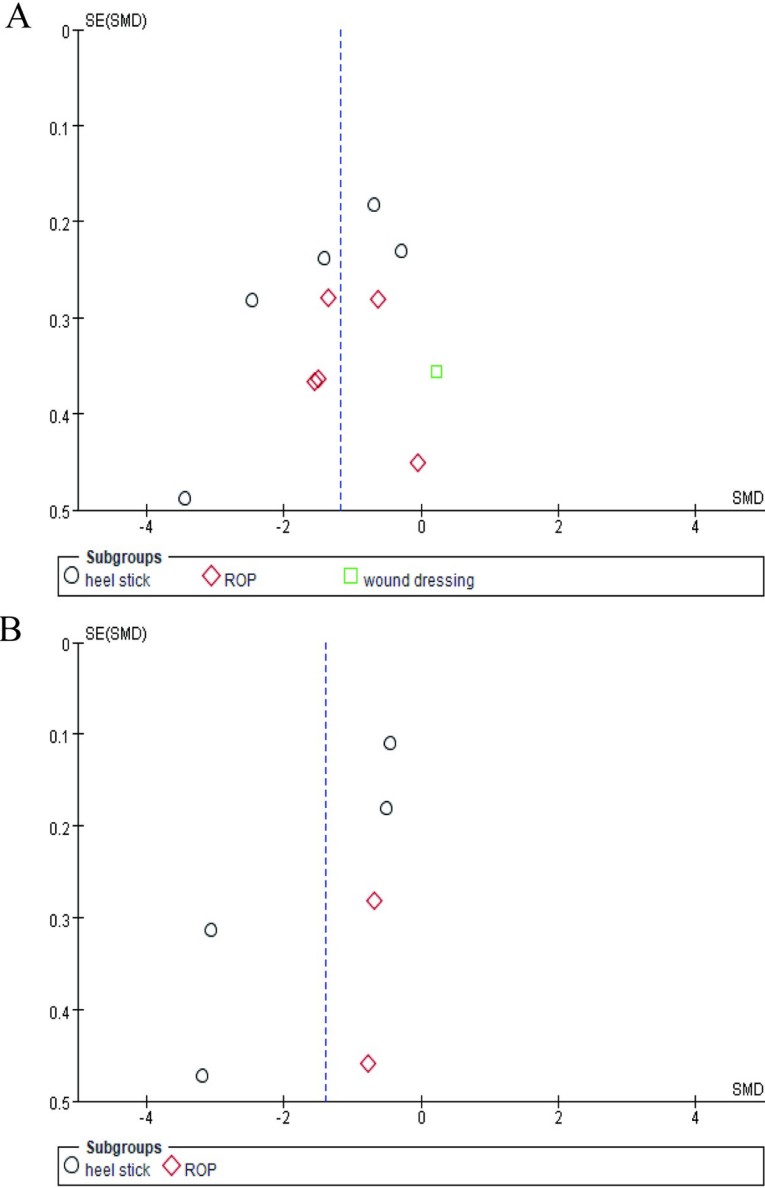

**Fig 9.** Funnel plots of publication bias on NNS+OS versus NNS alone (A) and NNS+OS versus OS alone (B) during painful procedures. NNS: nonnutritive sucking; OS: oral sucrose.

(intercept) was estimated -3.3 (95%CI -10.90, 4.30) with p value 0.3515 in the comparison between NNS+OS and NNS alone (11 studies). The estimates of bias were -5.60 (95%CI -13.40, 2.20) with p value 0.117 in the comparison between NNS+OS and OS alone (6 studies). The funnel plots presented slight asymmetry especially in heel-stick subgroup. More studies are needed to justify this publication bias.

### 3.3 Adverse events

Among the included studies, ten trials [36–38, 40, 43–45, 47–49] reported the occurrence of adverse events. Five trials observed that the occurrence rate of adverse events was zero in all treatment groups [36, 37, 43, 47, 48]. Oxygen desaturation was the most common adverse

**Table 2. Episodes of adverse events in included studies.**

| Study | NNS+sucrose: N(Total) | NNS alone: N(Total) | Sucrose alone: N (Total) | Routine care: N(Total) | Clinical significance |
|---|---|---|---|---|---|
| Dilli, 2014 [49] | bradycardia:6(32) | bradycardia:7 (32) | - | - | bradycardia: p = 0.75 |
|  | tachycardia:12(32) | tachycardia:19 (32) |  |  | tachycardia: p = 0.08 |
|  | desaturations:6(32) | desaturations:7 (32) |  |  | desaturations: p = 0.75 |
| Gao, 2018 [44] | vomit:1(22) | vomit:1(22) | - | Vomit:1 | vomit: $p = 0.800$; |
|  | abdominal distension:1 (22) |  |  | abdominal distension:1 | abdominal distension: $p = 0.562$ |
| Gibbins, 2002 [45] | No adverse events | oxygen desaturation:2 (64) | oxygen desaturation:3 (64) | - | p>0.05 |
|  |  | choke:1(64) |  |  |  |
| O'Sullivan, 2010 [38] | bradycardia:1(20) | bradycardia:3(20) | - | - | bradycardia: p = 0.328 |
|  | desaturations:1(20) | desaturations:3(20) |  |  | desaturations: p = 0.328 |
| Thakkar, 2016 [40] | oxygen desaturation:1(45) | oxygen desaturation:1 (45) | oxygen desaturation:1 (45) | oxygen desaturation:2 (45) | p>0.05 |

event, reported in 4 trials [38, 40, 45, 49]. Gao reported 5 episodes of vomit and abdominal distension whereas Dilli and O'Sullivan reported 13 and 4 episodes of bradycardia, respectively. A significant difference in tachycardia between the NNS+sucrose and the NNS alone group was observed by Dilli, whereas no significant adverse effects were observed in other trials (Table 2). Three trials reported heart rate and oxygen saturation as mean (SD) as a secondary outcome (Table 3) [34, 44, 46]. They all observed a significant effect of NNS+sucrose on oxygen saturation and heart rate except a non-significant result in Asmerom's for heart rate.

## 4 Discussion

Pain management has been well established in recent years, especially in procedures involving skin punctures [12, 50]. However, there is not a practical guideline on how to reduce pain and stress in a prolonged procedure such as ROP and wound treatment [51]. Therefore, this systematic review and meta-analysis aims to determine the effect of combined NNS and sucrose intervention in different types of painful procedures.

Overall, the combined interventions of NNS and sucrose showed the superiority in relieving mild pain during the painful procedures compared to applying NNS alone or sucrose alone or standard care. This has been confirmed by previous reviews [10, 13]. Non-nutritive sucking was believed to be associated with antinociceptive mechanisms and sucrose appears to

**Table 3. Summarized statistics of oxygen desaturation and Heart rate in included studies during painful procedures.**

| Study | Sucrose+NNS | | NNS | | sucrose | | Routine care | | p-value |
|---|---|---|---|---|---|---|---|---|---|
|  | Mean (SD) | N | Mean (SD) | N | Mean (SD) | N | Mean (SD) | N |  |
| oxygen saturation |  |  |  |  |  |  |  |  |  |
| Asmerom, 2013 [46] | 96.4(0.6) | 44 | 95.8(0.6) | 45 | - | - | - | - | <0.0001 |
| Gao, 2018 [44] | 95.2(1.6) | 22 | 92.9(2.4) | 22 | 93.5(1.7) | 21 | 92.9(2.1) | 21 | <0.05 |
| Miller, 2009 [34] | 97.69(2.41) | 14 | - | - | - | - | 94.30(3.74) | 14 | <0.001 |
| Heart rate |  |  |  |  |  |  |  |  |  |
| Asmerom, 2013 [46] | 170.5(14.7) | 44 | 164.9(14.6) | 45 | - | - | - | - | 0.07 |
| Gao, 2018 [44] | 138.6(7.9) | 22 | 154.2(9.0) | 22 | 151.6(9.6) | 21 | 156.8(7.2) | 21 | <0.0001 |
| Miller, 2009 [34] | 135.64(7.71) | 14 | - | - | - | - | 150.64(7.53) | 14 | <0.001 |

NNS: nonnutritive sucking; SD: Standard deviation.

enhance the effect of NNS, leading to an increase of endogenous endorphins [9, 52]. However, the mechanism of the combined effects is not completely clear [53].

On the other hand, the superiority is not significant when compared with direct breastfeeding or giving breast milk via a pacifier [36, 47]. The similarity between sucking through a pacifier and breastfeeding might explain this result [52]. Besides, swaddling and a blanket used in the two trials helped establish a simulated environment as breastfeeding, calming and comforting the babies. The similar effectiveness in healthy term neonates or stable late preterm neonates between sucrose/glucose administration and breast milk was also proposed in Shah's review [54].

ROP examinations and wound dressing have two points in common. Both procedures last for several minutes and tend to give giving stronger painful stimuli. The insertion of an eyelid speculum and scleral depression usually increases the intensity of the pain during ROP screening [55]. Although a significant effect was observed in ROP examinations in a preterm population, we found the means of PIPP score even in the combined NNS and sucrose group still around or higher than 12, qualitatively, indicating a moderate-to-severe level. Considering the preterm infants tend to have lower scores than term infants, such high scores indicated the finite effect of this combined intervention [56]. Besides, a non-significant effect was detected during wound dressing in Mandee's study where moderate-to-severe pain were often observed [37]. This might not be indicative because few studies researched on pain managements in wound dressing. However, previous studies found the effect of sucrose were time-dependent and could not last for a long time [10, 41]. This might be able to explain the minor effect. More innovations are needed on non-pharmacological interventions for those intensive procedures. For example, a recent study recommended the physiological flexion position called ROP position and it was demonstrated more effective than the combination of sucrose and NNS [57]. The combination of multiple non-pharmacological methods, including non-nutritive sucking, oral breast milk, and facilitated tucking, music, etc. might also help enhance the effects of pain relief [17, 18, 58]. Investigations on more effective methods are warranted, especially for lowering moderate or even severe pain.

Few studies in our analysis reported the pain measurements in the recovery phase, which might lead to an unreliable result in the meta-analysis. Our analysis found the effect has vanished in heel stick procedure when comparing the combined interventions with NNS alone group during this period. This finding could be reversed by removing a study with nonsignificant result in sensitivity analysis [39]. In this trial, both intervention and control groups received a pacifier and facilitated tucking resulting a low PIPP and sucrose did not contribute much to reducing pain. In fact, facilitated tucking has been reported to be effective in pain management [59].

Subgroup analysis on sucrose administration was not performed in the meta-analysis because of the varieties in different trials in terms of volumes (from 0.1 mL to 2 mL), concentrations (range from 12% to 33%) and frequencies. However, Steven found no difference in effectiveness among doses by proposing the minimal effective dose of only 0.1 mL in relieving procedural pain [60]. Thus, the wide range of volumes of sucrose might not influence the comparison.

OS has been considered to cause adverse events such as desaturation and bradycardia, especially in preterm infants [10]. Our analysis indeed indicates the trend of oxygen saturation increasing while heart rate decreasing under intervention. However, we did not observe a higher occurrence of bradycardia, tachycardia or desaturations in groups with OS. Besides, adverse events indeed occurred more in preterm subjects. Trials with zero adverse events were mostly based on full-term infants. This might indicate the vulnerability of preterm infants attributes more to adverse events than OS. Nonetheless, the evidence might be influenced by

reporting bias because some studies did not report any information on the safety during the intervention procedure.

Although the combined intervention of sucrose and NNS was considered effective in reducing pain among the studies included, sucrose was often administered 2 min prior to the painful procedure in these studies. This time interval was unjustified by Meesters in 2017 [61]. Besides, the recovery phase was neglected in half of the studies. The recovery from pain is important as it also affects the baby in a long-term way. We might improve the intervention methods to deal with moderate and severe pain and pay more attention to the recovery phase.

De Bernardo's trial found sucrose was more effective then oral glucose when both combined with a pacifier. This conflicts with a previous review which take glucose as an acceptable alternative to sucrose [62]. More studies could be conducted in this direction in order to expand the available and effective intervention methods.

There are three limitations in our analysis. Firstly, the heterogeneity is non-negligible in heel-stick subgroups but acceptable in other painful procedures. Heel-stick was involved in most studies. The variations in their protocols, such as population, the operating process, the pain measurements, or the sample size might interactively contribute to moderate to high heterogeneity. For example, some studies enrolled both preterm and term infants in the trial, while others enrolled purely preterm or full-term infants, which added the difficulty in explaining the effects on the certain population. Different scales and standards in pain measurements also influenced pooling data in quantitative analysis. We have conducted post hoc subgroup analyses to test the source of heterogeneity in terms of these factors. High heterogeneity might induce an unreliable conclusion even under a random-effects model. Results should be explained with caution. Secondly, pain score was the only one outcome to measure pain in this analysis and the different assessments were pooled. Those pain assessment methods are distinguishable from their target population, scales and evaluation items [63]. It was hard to unify because PIPP was the most common and other assessments accounted for no more than 25% of included studies. A sensitivity analysis was conducted to reduce the potential bias though. Other measurements including crying time were not investigated because of the limited number of studies. There were very few studies reporting crying time and comparing the combined intervention with sucrose alone or routine care. A single outcome of pain score might not be robust enough since different scales were used among studies. Finally, publication bias seems to exist in our analysis.

## 5 Conclusion

In conclusion, our systematic review and meta-analysis indicated the superiority of the combined intervention of sucrose and NNS than any single intervention except for breastfeeding. However, the effect appears to be mild in alleviating moderate-to-severe pain. More exploration and improvement of intervention were needed.

## Supporting information

**S1 Checklist. PRISMA 2009 checklist.**
(DOCX)

**S1 Table. Full searching strategy (from Jan 1, 2000 to Mar 31, 2021).**
(DOCX)

## Author Contributions

**Conceptualization:** Qiaohong Li, Xuerong Tan, Yongrong Zou.

**Data curation:** Qiaohong Li, Xuerong Tan, Xueqing Li, Wenxiu Tang, Lin Mei, Gang Cheng.

**Formal analysis:** Qiaohong Li, Xuerong Tan, Lin Mei, Gang Cheng.

**Funding acquisition:** Qiaohong Li, Yongrong Zou.

**Investigation:** Qiaohong Li, Xuerong Tan, Xueqing Li, Wenxiu Tang.

**Methodology:** Qiaohong Li, Xuerong Tan, Wenxiu Tang, Lin Mei.

**Project administration:** Qiaohong Li, Xuerong Tan, Xueqing Li, Yongrong Zou.

**Resources:** Qiaohong Li, Xueqing Li, Wenxiu Tang.

**Software:** Qiaohong Li, Xuerong Tan, Lin Mei, Gang Cheng.

**Supervision:** Qiaohong Li, Xueqing Li.

**Validation:** Qiaohong Li, Xuerong Tan, Xueqing Li, Wenxiu Tang, Gang Cheng.

**Visualization:** Qiaohong Li, Xuerong Tan.

**Writing – original draft:** Qiaohong Li, Xuerong Tan, Xueqing Li, Wenxiu Tang, Lin Mei.

**Writing – review & editing:** Qiaohong Li, Yongrong Zou.

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
