## [Decision Letter · Decision Letter 0]

27 Dec 2021

PONE-D-21-21132Efficacy and safety of combined oral sucrose and nonnutritive sucking in pain management for infants: A systematic review and meta-analysisPLOS ONE

Dear Dr. Zou,

Thank you for submitting your manuscript to PLOS ONE. After careful consideration, we feel that it has merit but does not fully meet PLOS ONE’s publication criteria as it currently stands. Therefore, we invite you to submit a revised version of the manuscript that addresses the points raised during the review process.

We look forward to receiving your revised manuscript.

Kind regards,

Girish Chandra Bhatt, MD, FASN

Academic Editor

PLOS ONE

Journal Requirements:

“Our paper is supported by Project of Sichuan Provincial Health Commission in 2014 (NO: 140108).”

One or more authors are affiliated with the funder, but authors state that the funder had no role

Thank you for stating the following financial disclosure:

We note that one or more of the authors is affiliated with the funding organization, indicating the funder may have had some role in the design, data collection, analysis or preparation of your manuscript for publication; in other words, the funder played an indirect role through the participation of the co-authors. If the funding organization did not play a role in the study design, data collection and analysis, decision to publish, or preparation of the manuscript and only provided financial support in the form of authors' salaries and/or research materials, please do the following:

a. Review your statements relating to the author contributions, and ensure you have specifically and accurately indicated the role(s) that these authors had in your study. These amendments should be made in the online form.

b. Confirm in your cover letter that you agree with the following statement, and we will change the online submission form on your behalf:

“The funder provided support in the form of salaries for authors [insert relevant initials], but did not have any additional role in the study design, data collection and analysis, decision to publish, or preparation of the manuscript. The specific roles of these authors are articulated in the ‘author contributions’ section.

4. PLOS requires an ORCID iD for the corresponding author in Editorial Manager on papers submitted after December 6th, 2016. Please ensure that you have an ORCID iD and that it is validated in Editorial Manager. To do this, go to ‘Update my Information’ (in the upper left-hand corner of the main menu), and click on the Fetch/Validate link next to the ORCID field. This will take you to the ORCID site and allow you to create a new iD or authenticate a pre-existing iD in Editorial Manager. Please see the following video for instructions on linking an ORCID iD to your Editorial Manager account: https://www.youtube.com/watch?v=_xcclfuvtxQ.

6. Please include a copy of Table 2 & 3 which you refer to in your text on page 13.

Reviewers' comments:

Reviewer's Responses to Questions

**Comments to the Author**

1. Is the manuscript technically sound, and do the data support the conclusions?

Reviewer #1: Yes

Reviewer #2: Partly

2. Has the statistical analysis been performed appropriately and rigorously? 

Reviewer #1: Yes

Reviewer #2: No

3. Have the authors made all data underlying the findings in their manuscript fully available?

Reviewer #1: Yes

Reviewer #2: Yes

4. Is the manuscript presented in an intelligible fashion and written in standard English?

Reviewer #1: Yes

Reviewer #2: No

5. Review Comments to the Author

Reviewer #1: Interesting, well written and well designed review and meta--analysis on efficacy and safety of non-pharmacological interventions on pain relief in infants.

Here my comments:

- introduction:once you introduce the abbreviation e.g OS, I suggest to use it instead of repeating oral sucrose.

- methods: please clarify if the research was restricted for english language articles or not; the statements were not clear for this point.

- discussion: you were discussiong about non pharamcological intervention; I think that citing studies about ketorolac was unnecessary.

Reviewer #2: The authors have made great efforts in doing this systematic review and meta-analysis. There are few concerns :

1. How was the cross over trial managed and which data was included in the meta-analysis is not mentioned. The methods section should clearly indicate how the cross-over designs will be dealt with.

2. The forests plots show critical levels of heterogeneity despite the subgroup analysis. The manuscript doesn't describe any efforts to explore the reasons for this heterogeneity. It is also worth considering if the meta-analysis with such high level of heterogeneity is required. If yes please add few lines about exploring heterogeneity. It is acceptable to have unexplained heterogeneity but a justification and reasoning with references is required.

3. Authors have interpreted the funnel plot as showing publication bias. It would be better to add Eggers test and then describe the funnel plots. Only consider funnel plots if number of studies in an analysis are more then 10

4. Authors have mentioned "sensitivity analysis conducted on 2 aspects..." please describe these 2 aspects and also add some details about sensitivity analysis in the method section

5. Overall a relook at the grammar and rephrasing of certain sentences is advised.

6. PLOS authors have the option to publish the peer review history of their article (what does this mean?). If published, this will include your full peer review and any attached files.

Reviewer #1: No

Reviewer #2: No

---

## [Author Response · Author response to Decision Letter 0]

21 Feb 2022

Response letter

Dear Editors and Reviewers,

 Thank you for your valuable comments. We have revised them one by one according to these comments. We hope that the revised manuscript is now acceptable for publication in your journal. 

 Thank you again for your time and consideration. 

Reviewers' comments:

Reviewer's Responses to Questions

Comments to the Author

1. Is the manuscript technically sound, and do the data support the conclusions?

Reviewer #1: Yes

Reviewer #2: Partly

2. Has the statistical analysis been performed appropriately and rigorously?

Reviewer #1: Yes

Reviewer #2: No

3. Have the authors made all data underlying the findings in their manuscript fully available?

Reviewer #1: Yes

Reviewer #2: Yes

4. Is the manuscript presented in an intelligible fashion and written in standard English?

Reviewer #1: Yes

Reviewer #2: No

5. Review Comments to the Author

Reviewer #1: Interesting, well written and well designed review and meta--analysis on efficacy and safety of non-pharmacological interventions on pain relief in infants.

Here my comments:

- introduction:once you introduce the abbreviation e.g OS, I suggest to use it instead of repeating oral sucrose.

Reply: Thanks. This problem has been updated.

- methods: please clarify if the research was restricted for english language articles or not; the statements were not clear for this point.

Reply: “Non-English literature were excluded.” was clarified in the exclusion criteria.(page 5 line 97)

- discussion: you were discussing about non pharamcological intervention; I think that citing studies about ketorolac was unnecessary.

Reply: Thanks for your suggestion. The references and related opinions have been deleted.

Reviewer #2: The authors have made great efforts in doing this systematic review and meta-analysis. There are few concerns :

1. How was the cross over trial managed and which data was included in the meta-analysis is not mentioned. The methods section should clearly indicate how the cross-over designs will be dealt with.

Reply: The results of cross-over design trials were standardized. The details were added to the method section. (page 7 line 130)

2. The forests plots show critical levels of heterogeneity despite the subgroup analysis. The manuscript doesn't describe any efforts to explore the reasons for this heterogeneity. It is also worth considering if the meta-analysis with such high level of heterogeneity is required. If yes please add few lines about exploring heterogeneity. It is acceptable to have unexplained heterogeneity but a justification and reasoning with references is required.

Reply: Possible explanations of high heterogeneity have been added to the limitation part. (page 17 line 342)

3. Authors have interpreted the funnel plot as showing publication bias. It would be better to add Eggers test and then describe the funnel plots. Only consider funnel plots if number of studies in an analysis are more then 10

Reply: Thanks for your advice. Egger’s Test has been added. (page 12 line 235)

4. Authors have mentioned "sensitivity analysis conducted on 2 aspects..." please describe these 2 aspects and also add some details about sensitivity analysis in the method section

Reply: Two aspects were described(page 10 line 192). Some sentences about sensitivity analysis were added to the last paragraph in method section. (page 7 line 148)

5. Overall a relook at the grammar and rephrasing of certain sentences is advised.

Reply: Thank you. The whole manuscript has been updated.

6. PLOS authors have the option to publish the peer review history of their article (what does this mean?). If published, this will include your full peer review and any attached files.

Do you want your identity to be public for this peer review? For information about this choice, including consent withdrawal, please see our Privacy Policy.

Reviewer #1: No

Reviewer #2: No

Finally, we would like to express my sincere appreciations to you for your hard work, dear editors and reviewers. We look forward to having more opportunities to learn from you in the future!

Best regards!

---

## [Editor Report · Decision Letter 1]

21 Apr 2022

Efficacy and safety of combined oral sucrose and nonnutritive sucking in pain management for infants: A systematic review and meta-analysis

PONE-D-21-21132R1

Dear Dr. Zou,

We’re pleased to inform you that your manuscript has been judged scientifically suitable for publication and will be formally accepted for publication once it meets all outstanding technical requirements.

Kind regards,

Girish Chandra Bhatt, MD, FASN

Academic Editor

PLOS ONE
---

## [Editor Report · Acceptance letter]

28 Apr 2022

PONE-D-21-21132R1 

Efficacy and safety of combined oral sucrose and nonnutritive sucking in pain management for infants: A systematic review and meta-analysis 

Dear Dr. Zou:

I'm pleased to inform you that your manuscript has been deemed suitable for publication in PLOS ONE. Congratulations! Your manuscript is now with our production department. 

Kind regards, 

on behalf of

Dr. Girish Chandra Bhatt 

Academic Editor

PLOS ONE